# Effect of Long-Term Adaptation to Cold and Short-Term Cooling on the Expression of the TRPM2 Ion Channel Gene in the Hypothalamus of Rats

**Anna A. Evtushenko *[iD], Irina P. Voronova and Tamara V. Kozyreva**

Department of Thermophysiology, Scientific Research Institute of Neurosciences and Medicine,
630117 Novosibirsk, Russia
* Correspondence: evtushenkoaa@neuronm.ru; Tel.: +7-913-767-72-68

**Abstract:** The present study is aimed to elucidate the possible involvement of the thermosensitive TRPM2 ion channel in changing of the temperature sensitivity of the hypothalamus after different cold exposures—long-term adaptation to cold and short-term cooling. Quantitative RT-PCR was used to study the expression of the gene of thermosensitive TRPM2 ion channel in the hypothalamus in the groups of control (kept for 5 weeks at +20 to +22 °C) and cold-adapted (5 weeks at +4 to +6 °C) rats, as well as in the groups of animals which were subjected to acute cooling (rapid or slow) with subsequent restoration of body temperature to the initial level. It has been shown that after long-term adaptation to cold, the decrease in the *Trpm2* gene expression was observed in the hypothalamus, while a short-term cooling does not affect the expression of the gene of this ion channel. Thus, long-term adaptation to cold results in the decrease in the activity not only of the TRPV3 ion channel gene, as shown earlier, but also of the *Trpm2* gene in the hypothalamus. The overlapping temperature ranges of the functioning of these ion channels and their unidirectional changes during the adaptation of the homoeothermic organism to cold suggest their functional interaction. The decrease in the *Trpm2* gene expression may indicate the participation of this ion channel in adaptive changes in hypothalamic thermosensitivity, but only as a result of long-term cold exposure and not of a short-term cooling. These processes occurring at the genomic level are one of the molecular mechanisms of the adaptive changes.

**Keywords:** TRP ion channels; TRPM2; gene expression; hypothalamus; long-term adaptation to cold; short-term cooling





## 1. Introduction

Temperature homeostasis is crucial to the functioning of a warm-blooded organism and requires precise regulation by the nervous system. More and more attention is being paid to the study of molecular mechanisms of temperature sensitivity. An important milestone in this direction was the discovery of thermosensitive TRP (transient receptor potential) ion channels: cold-sensitive TRPM8 and TRPA1, as well as heat-sensitive TRPV1, TRPV2, TRPV3, and TRPV4 [1–3]. These temperature-activated TRP ion channels cover the temperature range perceived by most mammals [4,5]. All of them were found both in the peripheral nervous system and in the central structures of the brain [6–11], including the hypothalamus [6,10–13]. Currently, these TRP ion channels are considered as primary detectors of temperature changes at the periphery [4,5,14] and as it was supposed by McKemy [15] and Caterina [4] at the level of central structures.

Recently, the thermosensitive TRPM2 ion channel has attracted the attention of researchers [14,16–20]. Although in 2006 Togashi et al. showed the ability of TRPM2 to be activated when exposed to a temperature in the range of 34–38 °C with a high $Q_{10}$ value of about 15.6 [21], TRPM2 was not classified as thermosensitive, since there were no data on its presence in sensory neurons. Later, Naziroglu et al. [22] and then Tan and McNaughton [17]

found TRPM2 expression in 10% of neurons of dorsal root ganglia (DRG) and in 85% of heat-sensitive neurons in cultures of DRG. To date, it has been shown that TRPM2 is widely represented in various organs and tissues, including the brain [18,23–25].

Works aimed at elucidating the physiological role of TRPM2 in maintaining temperature homeostasis in a warm-blooded organism indicate that it can be involved in the mechanisms of temperature perception both at the periphery [26] and in the center [18–20]. This suggests the possible involvement of the TRPM2 ion channel in providing temperature sensitivity of hypothalamic neurons.

While studying the expression of genes of thermosensitive TRP ion channels in the hypothalamus, we found the decrease in the level of expression of the TRPV3 ion channel gene in the cold-adapted rats by 30% compared with the control animals. The level of gene expression of other studied thermosensitive TRP ion channels (TRPV1, TRPV2, TRPV4, TRPM8, and TRPA1) did not change significantly [10,11]. Acute short-term cooling with subsequent restoration of body temperature to the initial level resulted in a twofold increase in the expression of the TRPM8 ion channel gene [27]. The gene encoding the TRPM2 ion channel was not studied in these experiments.

The aim of the present study was to evaluate the expression of the thermosensitive TRPM2 ion channel gene in the center of thermoregulation–hypothalamus, after different cold exposures on a warm-blooded organism: long-term adaptation to cold and short-term (rapid or slow) cooling, followed by the restoration of body temperature to the initial level.

## 2. Materials and Methods

### 2.1. Animals

Males of Wistar rats were used throughout. Animals were obtained from the Institute of Cytology and Genetics (Siberian Branch of the Russian Academy of Sciences, Novosibirsk) and weighed 250–300 g. Initially, animals were housed in groups of 5 per cage at an ambient temperature of +20 to +22 °C, on a 12 h dark/light cycle, with free access to feed and water. Two series of experiments were performed: long-term adaptation to cold and acute cooling (rapid or slow) with subsequent restoration of body temperature to the initial level. All experimental procedures were in compliance with Directive 2010/63/EU of the European Parliament and were approved by the ethics committee of the Scientific Research Institute of Neurosciences and Medicine (Novosibirsk, Russia).

### 2.2. Long-Term Adaptation to the Cold

Thermal adaptation was performed according to Hart's protocol [28]: animals were housed for 5 weeks in individual cages 60 × 40 × 20 cm under a 12:12 h light–dark cycle (light on at 8.00 AM) with free access to food and water in temperature-regulated rooms. The temperatures of rooms were maintained at +4 to +6 °C for cold-adapted rats and at +20 to +22 °C for non-adapted (control) rats. After 5 weeks of adaptation to cold, the cold-adapted (n = 10) and control (n = 10) rats were decapitated, their brains were rapidly removed on ice, and samples of the hypothalamus were taken for PCR analysis.

### 2.3. Acute Cooling with Subsequent Restoration of Body Temperature to the Initial Level

2.3.1. Preparation of Animals

Three to four days before the experiment, the rats were isolated by placement in separate cages to eliminate the group effect. Experiments were performed on anesthetized animals (nembutal 40 mg/kg initially and 20 mg/kg 35–40 min after the first injection) to exclude the effects of movements and of emotional stress. Before experiment, the abdomen skin area of 25 cm$^2$ was freed from hair. The removal of hair was carried out by scissors without any chemical drugs and painful irritation. At the middle of this area, an intradermal thermosensor was placed. Moreover, a rectal thermosensor lubricated by mineral oil was inserted in its place (at depth of 6 cm) and fixed. During these manipulations, rats were on a warm table to avoid an uncontrolled body temperature decrease in anesthetized animals. Rats prepared for experiment were restrained on the temperature-regulated table.

### 2.3.2. Cold Exposure

The experiments were conducted at ambient temperature at +20 to +22 °C. The cooling was implemented by means of a thermode with an area of 25 cm², applied to the abdominal skin where hair was shaved off beforehand. Models of rapid and slow cooling were used. These models differ as follows: there is dynamic activity of cutaneous thermoreceptors during the rapid cooling, but there is no such activity during the slow cooling [29,30], as well as the order of thermoregulatory responses is somewhat different [31]. Moreover, the thermoregulatory responses for the second cooling depend on the rate of the first cooling [32]. The rate of the skin temperature decrease during the rapid cooling was 0.06 ± 0.004 °C/s, whereas during the slow cooling it was 0.006 ± 0.0005 °C/s. In all cases, the depth of cooling was the same: a decrease in rectal temperature by 3 °C.

### 2.3.3. Registered Temperature Parameters

In rats, the control of core body temperature was realized by a rectal thermosensor. The rate of cooling was controlled by the intradermal thermosensor. These parameters were registered continuously with an MP100 system (set of instruments) (BIOPAC Systems Inc., Goleta, CA, USA). Data preprocessing was performed in the AcqKnowlege software supplied with this system.

### 2.3.4. Design of the Experiment

Each anesthetized animal was restrained on a thermostated table and rested there until the stabilization of rectal and intradermal temperatures. After that it was subjected to cooling (14.0 ± 1.1 min in the rapid cooling model; 25.4 ± 1.9 min in the slow cooling model), warming up to initial body temperature (101.1 ± 3.1 min in the rapid cooling group; 105.2 ± 2.9 min in the slow cooling group) and decapitation. Control animals (not exposed to the cold) rested on the warm thermostated table during 106.4 ± 2.2 min and decapitated at the end of this period.

### 2.3.5. Experimental and Control Groups of Animals

Three experimental groups of rats were set up: group 1—animals subjected to rapid cooling and subsequent restoration of body temperature to the initial level; group 2—animals subjected to slow cooling and subsequent restoration of body temperature to the initial level; and group 3—control animals. There were 10 animals in each group.

### 2.4. Collection of the Samples for PCR Analysis

After the decapitation, rat brains were rapidly removed on ice and the hypothalamic area including the *tuber cinereum* with its preoptic part and mamillary bodies were excised at approximately 1.5 mm depth (the *chiasma opticus* and *hypophysis* were removed previously). The tissue samples were placed in sterile tubes, frozen under liquid nitrogen, and stored at −70 °C until total RNA extraction.

### 2.5. Quantitative RT-PCR

Total RNA was extracted with TRIzol Reagent (Thermo Fisher Scientific, Waltham, MA, USA) according to the manufacturer's protocol. Possible traces of genome DNA were removed by RQ1 RNase-Free DNase (Promega, Madison, WI, USA) treatment according to the manufacturer's protocol. RNA solution was diluted with DEPC-treated water to the concentration of 0.15 µg/µL and stored at −70 °C until use. The Reverse Transcription M-MuLV–RH reagent kit (Biolabmix, Novosibirsk, Russia) with random hexanucleotide primer was used for cDNA synthesis. The absence of contamination with genomic DNA in the cDNA samples was checked by PCR with primers for tryptophan-hydroxylase I (*Tph1*; this gene is not expressed in the brain) at the 36 cycles [33,34]. All the primers utilized in this study were composed on the base of the sequences published in the European Molecular Biology Laboratory (EMBL) nucleotide database and were synthesized in the Biosset Company (Novosibirsk, Russia). The PCR product of each primer pair was analyzed in

agarose gel electrophoresis to verify the quality and length of produced amplicons. Primer characteristics are given in Table 1.

**Table 1.** Characteristics of primers used to quantify mRNA of thermosensitive TRPM2 ion channel in rat hypothalamus.

| Gene | Primer Sequence | Annealing Temperature, °C | Number of Cycles | Amplicon Size, Bp. |
|---|---|---|---|---|
| *Trpm2* | F 5′-GACAGCAACCACTCCCACTT<br>R 5′-CTCCAACACCACGCAGACA | 60 | 45 | 162 |
| *Ppia* | F 5′-TTCCAGGATTCATGTGCCAG<br>R 5′-CTTGCCATCCAGCCACTC | 64 | 32 | 206 |
| *Tph1* | F 5′-**GAAAGTATTTCGCAGAGCTGG**<br>R 5′-**GGCGTGGGTTGGGTAGAGTTTGTT** | 66 | 36 | 283 with intron,<br>134 without intron |
| **Intraexonic Primers** | | | | |
| *Trpm2* | F 5′-**CAACCACTCCCACTTCATCCT**<br>R 5′-**CCCTCTTTCCTTCGTTTGCTCC** | 60 | 32 | 115 |
| *Ppia* | F 5′-CCGACTGTGGACAACTCTAAT<br>R 5′-ACTTGAAGGGGAATGAGGAAA | 61.5 | 25 | 168 |

qPCR was carried out on a LightCycler-480 II thermocycler (Roche, Basel, Switzerland). The reaction mixture contained a master mix with SYBR Green (BioMaster HS-qPCR SYBR Blue(2x), Biolabmix, Novosibirsk, Russia), forward and reverse primers, and a cDNA template.

Protocol of amplification was as follows: initial denaturation of cDNA and activation of the enzyme for 5 min at 95 °C, N cycles (Table 1) consisting of denaturation at 95 °C for 10 s, primer annealing at specific temperature (Table 1) for 30 s, and elongation at 72 °C for 15 s with fluorescence signal acquisition. At the end of amplification, the melting curve was generated in the range of 72 °C to 95 °C for verification of reaction specificity. The standard-curve quantitation method was applied [35]: the relative amount of a tested cDNA was determined using calibration curves derived from the dilutions of a standard cDNA (1:1, 1:2, 1:4, 1:8, 1:16, 1:32, 1:64, and 1:128). The standard cDNA solution for plotting the calibration curves was prepared by mixing aliquots from each synthesized-cDNA sample. LightCycler-480 (Roche, Basel, Switzerland) software was used to build the calibration curves and calculation of the relative amount of cDNA in the samples. In order to obtain results as a number of copies of target gene per 100 copies of reference gene, we performed the quantification of a number of copies of target and reference genes in standard cDNA solution using the genomic DNA of known concentration as the external standard [36,37] and intra–exonic primers (Table 1). We used 0.5 ng/µL, 1 ng/µL, 2 ng/µL, 3 ng/µL, 4 ng/µL, 6 ng/µL, and 8 ng/µL of DNA as standard for *Trpm2*; and 1 ng/µL, 2 ng/µL, 4 ng/µL, 6 ng/µL, 8 ng/µL, 10 ng/µL, 12 ng/µL, 14 ng/µL, and 20 ng/µL of DNA as standard for *Ppia* (1 ng of genomic DNA contains about 200 copies of single-copy gene). Each value needed for characterizing the gene expression (relative concentrations of target and reference genes, as well as absolute amount of copies of target and reference genes in standard cDNA) was obtained in three repetitions.

The expression level of the TRPM2 ion channel gene is presented as a number of the TRPM2 ion channel gene mRNA copies per 100 copies of mRNA of a housekeeping gene (peptidyl-prolyl cis-trans isomerase A; *Ppia*), i.e., in copies/100 *Ppia* copies. The genomic *Tph1* fragment was undetectable in the samples used in the present study.

### 2.6. Statistics

Statistica 8 software package (StatSoft Russia) and Microsoft Excel software were employed for statistical analysis. The data are presented as mean ± SEM and were subjected to Student's t-test for comparison of identical parameters in control and experimental

groups, and to multivariate analysis of variance (ANOVA) followed by Fisher's post hoc test for multi-group comparisons. A difference was considered significant at $p < 0.05$.

## 3. Results

### 3.1. Long-Term Adaptation to the Cold

No differences ($p > 0.05$, $p = 0.73$, t = $-0.35$) in the level of the *Ppia* mRNA in the hypothalamus were found between the control ($0.56 \pm 0.05$ ng/µL) and cold-adapted ($0.58 \pm 0.05$ ng/µL) rats. This allowed us to use the *Ppia* mRNA level as an endogenous standard for comparing the data of these experimental groups of animals.

Studies of the gene expression of the thermosensitive TRPM2 ion channel in the hypothalamus of the control and cold-adapted rats showed that long-term cold adaptation resulted in a significant decrease (by 30%) in the *Trpm2* gene expression from $45.45 \pm 4.80$ to $31.61 \pm 4.12$ copies/100 copies of *Ppia* ($p < 0.05$, $p = 0.044$, t = 2.17) (Figure 1).

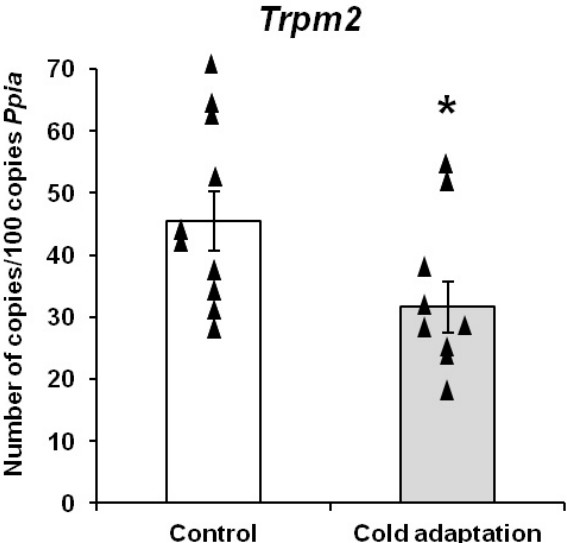

**Figure 1.** Effect of long-term adaptation to cold on expression of the *Trpm2* gene in the hypothalamus of rats. Significant differences between the control and cold-adapted rats: * $p < 0.05$ ($p = 0.044$, t = 2.17). Triangles represent the individual values.

### 3.2. Acute Cooling with Subsequent Restoration of Body Temperature to the Initial Level

There were no differences in the level of the *Ppia* mRNA in the hypothalamus of the rats in the control group without cooling ($3.13 \pm 0.14$ ng/µL) and that of the rats with rapid ($3.07 \pm 0.12$ ng/µL) and slow cooling ($3.04 \pm 0.17$ ng/µL) ($F_{(2,28)} = 0.09$, $p = 0.91$). Therefore, we used the *Ppia* mRNA level as an endogenous standard for these experimental groups of rats.

The levels of expression of the *Trpm2* gene in the hypothalamus in the control as well as in rapid- and slow-cooled animals were $61.88 \pm 9.93$, $67.54 \pm 10.15$, and $71.24 \pm 15.65$ copies/100 copies of *Ppia,* respectively. There were no significant differences in the expression of the thermosensitive TRPM2 ion channel gene in the hypothalamus of the rats after acute (rapid and slow) cooling followed by the restoration of body temperature to the initial level ($F_{(2,27)} = 0.14$, $p = 0.87$).

## 4. Discussion

The present study is aimed at elucidating the possible involvement of the TRPM2 ion channel in adaptive changes in the temperature sensitivity of hypothalamic neurons after different cold exposures on a warm-blooded organism.

The following temperature effects were used: a model of long-term adaptation to cold [28] and a model of a short-term (rapid or slow) cooling of the animal with subsequent restoration of body temperature to the initial level. The model of long-term adaptation

to cold [28] is a recognized model which we used for studying the physiology of thermoregulation [38], including at the genomic level [10,11,39]. Long-term adaptation of the organism to cold leads to significant physiological changes, bringing the body to a new level of functioning to maintain a deep body temperature. The body temperature of the rats kept at +4 to +6 °C (after 2 weeks of adaptation) is maintained at the same level as the body temperature of the control animals kept at +20 to +23 °C (Figure S1).

We chose the model of a short-term (rapid or slow) cooling of the animal with subsequent restoration of body temperature to the initial level based on the data of Brück, Zeisberger [40] and Kozyreva et al. [32]. In the experiments of these authors, the thermoregulation responses to a repeated cold stimulus after cooling of the animal and its subsequent warming up to the initial body temperature developed differently from the preceding cooling. This makes it possible to assume the presence of changes in the center of thermoregulation after the first presentation of a cold stimulus. In our study, we tried to determine whether these changes affect the thermosensitive TRPM2 ion channel at the level of gene expression.

We found that long-term and acute short-term cooling had different effects on the expression of the thermosensitive TRPM2 ion channel gene in the hypothalamus. Long-term adaptation to cold leads to a decrease in the expression of the TRPM2 ion channel gene that may indicate its involvement in adaptive changes in the thermosensitivity of hypothalamic neurons. A short-term (neither rapid nor slow) cooling causes no changes in the expression of the *Trpm2* gene in the hypothalamus.

Phylogenetically, TRPM2, being a member of the TRPM family, is close to the cold-sensitive TRPM8 ion channel [41]; however, no changes in the activity of this channel were detected in response to a cold stimulus [21]. Its activity increased with temperature stimuli above 34 °C, but it began to decrease at temperatures above 37 °C. This allowed the authors to suggest that the optimal temperature for TRPM2 activation was close to the internal temperature of a warm-blooded organism. It is noteworthy that the activity range of the TRPM2 ion channel (34–38 °C) practically coincides with the activity range of another thermosensitive ion channel, TRPV3 (31–39 °C) [2].

In our previous studies, it was shown that during long-term adaptation of rats to cold, the proportion of neurons sensitive to the temperature in the range of 35–38 °C decreased in the hypothalamus, but the proportion of neurons sensitive in the range of 38–41 °C increased [42] (Figure S2). Determination of the expression of thermosensitive ion channel genes (TRPV1, TRPV2, TRPV3, TRPV4, TRPA1, and TRPM8) in the hypothalamus showed a 30% decrease in the expression of the TRPV3 ion channel gene during long-term adaptation to cold compared with the control [10,11]. Since TRPV3 is activated at the temperature of 31–39 °C, it has been suggested that this ion channel may be responsible for providing the thermosensitivity of hypothalamic neurons that are sensitive to the temperature in the range of 35–38 °C and for their changes that occur during long-term adaptation to cold.

In the present study we found the 30% decrease in the expression of the TRPM2 ion channel gene in the hypothalamus of cold-adapted animals. The temperature activation ranges of the TRPV3 (31–39 °C) and TRPM2 (34–38 °C) ion channels are a practical coincidence. These facts suggest that TRPM2 is involved on par with TRPV3 in adaptive changes in the thermosensitivity of hypothalamic neurons during long-term cold adaptation of a warm-blooded organism. Undoubtedly, the TRPM2 and TRPV3 ion channels may be involved in the hypothalamic thermosensitivity mechanisms in different ways. This fact definitely requires experimental confirmation.

The decrease in the expression of the *Trpm2* gene (and, presumably, of the TRPM2 ion channel itself) found in this study as a result of long-term exposure to cold seems quite logical. It explains the decrease in the number of hypothalamic neurons sensitive in the temperature range of 35–38 °C in cold-adapted animals and agrees with the available data on the physiological role of TRPM2. Thus, according to Song et al. [18], activation of TRPM2-expressing POA neurons induces hypothermia. With long-term exposure to cold, the reactions of the body are rearranged to ensure normal body temperature. Therefore,

a decrease in the number of the ion channels contributing to hypothermia seems quite justified. It is also known that it is easier to induce a state of overheating in cold-adapted animals [43]. According to Kamm and Siemens, the physiological role of TRPM2 in the hypothalamus is to prevent an increase in body temperature. The authors made this conclusion based on a comparison of thermoregulatory responses in wild-type mice and *Trpm2*-knockout mice. The decrease in core body temperature in response to local heating of the hypothalamus is less pronounced in the knockout mice [19,20]. In addition, these mice develop a more pronounced febrile response compared with wild-type mice after pyrogen injections [18]. That is why, it is possible that the detected decrease in the expression of the *Trpm2* gene (and, presumably, of the TRPM2 ion channel itself) in cold-adapted animals underlies the occurrence of overheating processes described by Cabanac [43] in cold-adapted animals, and is probably the molecular mechanism of the initial link in the regulatory chain of these processes.

As for the TRPV3 ion channel, its involvement in the mechanisms of thermosensitivity both at the periphery and in the center is not well understood, and the data available to date are contradictory [44–46]. In addition to our results [10,11,27,47], we have not found data on the involvement of TRPV3 in the mechanisms of thermosensitivity in the hypothalamus in the literature. In any case, the question of the involvement of thermosensitive TRP ion channels in the mechanisms of maintaining temperature homeostasis and the role of each of them in these mechanisms requires further study.

The formation of adaptive changes begins just after the first presentation of a temperature stimulus and appears in the modified structure of the thermoregulatory response of the organism to similar repeated cooling [32,40,48,49]. The change in thermoregulatory responses after a short-term cooling indicates a change in the regulatory characteristics of the temperature homeostasis system [38]. In the center of thermoregulation, the hypothalamus, this may be the result of the change in the number of thermosensitive TRP ion channels embedded in the neuronal membrane, including the TRPM2 ion channel. However, in contrast to long-term exposure to temperature (cold adaptation), a short-term (rapid or slow) cooling with subsequent restoration of body temperature to the initial level did not affect the expression of the TRPM2 ion channel gene in the hypothalamus. In physiological experiments [32], in response to repeated cooling a change in the temperature thresholds of effector responses of animals was observed. This may indicate a change in the properties of central and peripheral thermoreceptors [32]. In our experiments, we took samples for analysis at the moment corresponding to the initiation of repeated cooling. However, as our study showed, TRPM2 of the hypothalamus does not take part in these processes (at least at the genomic level). We obtained a similar result earlier for the TRPV3 ion channel [27]. However, a twofold increase in the expression of the TRPM8 ion channel gene was registered [27], which suggests that this channel is involved in thermoregulatory rearrangements after a short-term cooling.

## 5. Conclusions

Thus, long-term adaptation to cold results in the decrease in the activity of not only the TRPV3 ion channel gene, as shown earlier, but also of the *Trpm2* gene in the hypothalamus. The overlapping temperature ranges of the functioning of these ion channels and unidirectional changes of their gene expression during the adaptation of the homeothermic organism to cold suggest their functional interaction. The decrease in the *Trpm2* gene expression may indicate the participation of this ion channel in adaptive changes in the hypothalamic thermosensitivity, but only as a result of long-term cold exposure and not of a short-term cooling. These processes occurring at the genomic level are one of the molecular mechanisms of the adaptive changes.

**Supplementary Materials:** The following supporting information can be downloaded at: https://www.mdpi.com/article/10.3390/cimb45020065/s1.

**Author Contributions:** T.V.K.—conception and design; A.A.E. and I.P.V.—acquisition of data and analysis, interpretation of data, drafting the manuscript; T.V.K.—revising the manuscript critically for important intellectual content and final approval of the version to be published. All authors have read and agreed to the published version of the manuscript.

**Funding:** This research received no external funding.

**Institutional Review Board Statement:** All experimental procedures were in compliance with the Directive 2010/63/EU of the European Parliament and were approved by the ethics committee of the Scientific Research Institute of Neurosciences and Medicine (Novosibirsk, Russia). Protocol No. 3-O, dated 18 March 2021.

**Informed Consent Statement:** Not applicable.

**Data Availability Statement:** All data are in the work logs (Department Archive).

**Acknowledgments:** The study was supported by budgetary funding for basic scientific research (theme No 122042700001-9).

**Conflicts of Interest:** The authors declare no conflict of interest.

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
