# Peer review of "Effect of Long-Term Adaptation to Cold and Short-Term Cooling on the Expression of the TRPM2 Ion Channel Gene in the Hypothalamus of Rats"

_cimb, doi:10.3390/cimb45020065_

Round 1
Reviewer 1 Report
The focus of the present manuscript was to quantify the amount of Trpm2 mRNA in hypothalamus of rats adapted to cold conditions, acute cooling or rats maintained at room temperature.
Overall the amount of data included in the manuscript is very low and only consists of real time PCR estimates of the Trpm2. The estimated amount of Trpm2 mRNA is shown to decrease 30% after cold-adaption but not to acute cooling. The authors discuss this very extensively anticipating that their mRNA data also reflects activity. A linear correlation between mRNA amount, protein amount and activity can however, nor be taken for granted. I therefore believe that the authors should investigate whether the accumulated amount of TRPM2 protein is decreased proportionally to the mRNA and also how temperature affects that activity.
The manuscript would furthermore benefit from English editing as the manuscript contains a number of sentences that are difficult to comprehend. I will however not go into detail with language issues as I believe that there are too little information in the present manuscript to justify publication in its present form.
Reviewer 2 Report
The present study is aimed to elucidate the possible involvement of the 9 thermosensitive TRPM2 ion channel in changing of the temperature sensitivity of the 10 hypothalamus after different short and long- germ cold exposures. Thermosensitive TRM as well TRPV series, can be involved in the mechanisms of temperature perception both at the periphery, and seems that decrease in the level of expression of the TRPV3 ion channel gene in the cold-adapted rats by 30% compared with the control animals.
There are some minor concerns:
Line 94: Nembutal does modify the rats temperature? Has this aspect been taken into consideration?
Line 202: I valori riportati sono al limite della significatività statistica e quindi necessitano di una ulteriore conferma con parametri numericamente più certi.
Lines 246 and following: “Long-term adaptation to cold leads to a decrease in the expression of the TRPM2 ion 246 channel gene that may indicate its involvement in adaptive changes in the 247 thermosensitivity of hypothalamic neurons. “…”Its activity increased with temperature stimuli above 34°C, but it began to decrease at temperatures above 37°C. This allowed the authors to suggest that the optimal temperature for TRPM2 activation was close to the internal temperature of a warm-blooded organism. It is noteworthy that the activity range of the TRPM2 ion channel (34–38°С) practically coincides with the activity range of another thermosensitive ion channel, TRPV3 (31–39°С) [2]”. This concept is very important, but it refers only to an evaluation of an experimental model and does not necessarily indicate a reproducible modality in other animal models or humans. This aspect must be emphasized.
Lines 275-282: repetitive, eliminate this sentence.
Lines 331 and following: This sentence should be placed in a "conclusions" section. - The discussion section is too long and could be shortened
Round 2
Reviewer 1 Report
The revised manuscript has improved some of the issues I raised during my first review. My basic problem is that I still believe, that more experiments must be included in the manuscript before it can be published. The authors address a number of these experiments in their rebutal letter and state that these should be the focus of future studies. I, however, believe these should be performed and included in the present manuscript before it can be accepted.